# Hypercoagulable State in COPD-A Comprehensive Literature Review

**DOI:** 10.3390/diagnostics11081447

**Published:** 2021-08-10

**Authors:** Christos Kyriakopoulos, Athena Gogali, Konstantinos Kostikas, Athanasios Konstantinidis

**Affiliations:** Respiratory Medicine Department, Faculty of Medicine, University of Ioannina, 45500 Ioannina, Greece; ckyriako123@gmail.com (C.K.); athenagogali@yahoo.com (A.G.); akonstan@uoi.gr (A.K.)

**Keywords:** hypercoagulability, coagulation factors, stable COPD, acute exacerbation COPD

## Abstract

Chronic Obstructive Pulmonary Disease (COPD) is a chronic inflammatory disease with multisystemic manifestations. Studies either held on stable disease patients or during exacerbations have demonstrated that COPD is strongly related to venous thromboembolism and cardiovascular events. The aim of the present review of the literature was to provide an in-depth overview regarding the alterations of coagulation factors and prothrombotic changes generated in patients with stable COPD and during COPD exacerbations.

## 1. Introduction

Chronic Obstructive Pulmonary Disease (COPD) is a chronic debilitating lung disease with a high prevalence of approximately 380 million cases worldwide [1]. It is currently the third leading cause of death, responsible for approximately 6% of the world’s total deaths (approximately 3.3 million annually) [2]. In addition to the known devastating respiratory consequences, a large number of studies support the hypothesis that COPD increases the risk for both venous thromboembolism (VTE) and cardiovascular disease (CVD). Thus, patients with COPD exhibit an enhanced risk of pulmonary embolism and deep vein thrombosis compared to randomly matched COPD-free subjects [3], and are also more likely to be diagnosed with cardiovascular disease compared with non-COPD subjects, having nearly 2.5 times the risk of CVD [4]. Although the observed association of COPD with CVD and VTE can be partially explained by comorbidities and shared risk factors, there is strong evidence that COPD increases the risk for cardiovascular morbidity and mortality independently of age, gender, and smoking history [5,6]. It is of note that up to 63.5% of patients with COPD die of comorbid circulatory system diseases [7].

Increased thrombin formation [8], reflected by elevated thrombin–antithrombin complexes [9], tissue factor procoagulant activity [10] and activated factor XI [11], increased d-dimers [12], and FI [13], FII and FX [14] levels in the serum of COPD patients support the theory that a hypercoagulable state occurs in patients with COPD and might contribute to the incidence of atherothrombotic events and VTE, increasing disease related morbidity and mortality. Pulmonary embolism accounts for approximately 10% of deaths in patients with stable COPD, on long term oxygen therapy (LTOT) [15]. Kim et al., demonstrated that BMI, exercise capacity, and medical comorbidities were significantly associated with VTE in moderate to severe COPD and that physicians should suspect VTE in patients who present with dyspnea and should consider possibilities other than infection as causes of COPD exacerbation [16].

Potential pathways illustrating pathogenetic mechanisms of increased risk of CVD and VTE in COPD are imprecise. Evidence illustrates four possible synergistic mechanisms: systemic inflammation [17], platelet activation [18], oxidative stress [19], and hypoxia, either sustained in severe COPD or intermittent during exercise and sleep [20,21]. COPD is a disease where both local and systemic inflammation is present [17]. Prolonged systemic inflammation in COPD is indicated by increased serum concentrations of inflammatory markers such as fibrinogen, C-reactive protein (CRP), interleukin 6, interleukin 8, and TNF-a [22]. The inflammatory response and the activation of coagulation are two important mechanisms of a host’s defense response against infection that do not work independently, but conjointly in an intricate and synchronous process [23]. Several investigators have suggested that low-grade inflammation in COPD largely contributes to a prothrombotic state [24]. Platelets serve a crucial role in hemostasis and inflammation [18]. Activated platelets produce several prothrombotic factors that also act as inflammatory signals, such as platelet factor 4, plasminogen activator inhibitor (PAI-1), von Willebrand factor (vWF), and fibrinogen [25]. Platelets also secrete soluble *P*-selectin and CD40L, which serve a dual purpose of promoting thrombus formationand activating inflammatory cells [23]. Platelet *P*-selectin has been shown to bind PSGL-1 on leukocytes to promote fibrin degradation and tissue factor release.

Oxidative stress is triggered by reactive oxygen species, predisposing to thrombosis by impairing red blood cells’ quality and function [26], inducing endothelial dysfunction and injury, and activating leukocytes, consequently affecting the clotting system [19,27]. Oxidative stress also elevates reactive oxygen species within platelets that can augment platelet activation and thrombotic susceptibility [18]. Mechanisms by which hypoxia can stimulate a prothrombotic response include platelet activation and hypoxia-inducible transcription factors (HIFs) mediated or HIF-independent increase in procoagulant factors or reduction of coagulation inhibitors [28]. Hypoxia can also induce PAI-1 in a time-dependent process; by enhancing TF and repressing tissue factor pathway inhibitor (TFPI) transcript expression [29], decreasing protein S levels and increasing thrombin generation, it contributes to the prothrombotic effect [30].

COPD patients are at heightened risk of VTE during acute exacerbations of COPD (AECOPD) [31,32], which may be due to clotting system activation [33]. AECOPDs, which are crucial events in the disease’s natural history, can concomitantly lead to hypoxia and amplification of systemic inflammation, thus magnifying the prothrombotic state [34]. Respiratory tract infections, the most frequent trigger of AECOPD, represent a transient risk factor for VTE [35]. In a population-based case-control study in Northern Denmark, the Incidence Rate Ratio (IRR) for VTE within the first 3 months after infection was 12.5 (95% CI: 11.3–13.9) for patients with hospital-diagnosed infection and 4.0 (95% CI: 3.8–4.1) for patients treated with antibiotics in the community, compared with individuals without infection during the year before VTE [36]. Immobilization is another factor, existent during AECOPD, which imposes an even greater risk for VTE [37,38].

Obstructive sleep apnea (OSA) is a frequently encountered comorbidity among COPD patients. The estimated prevalence of sleep disturbance in COPD varies from 34% to 78% [39,40]. OSA further magnifies the risk of VTE through hypoxia, systemic inflammation, endothelial damage, and connection to metabolic syndrome [41,42]. Xie et al., demonstrated that patients with COPD and OSA had higher odds of PE compared to control patients (individuals without OSA or COPD), with significance persisting after adjusting for covariates (OR 5.66; 95%CI 1.80–16.18, *p* = 0.004). Interestingly, patients with COPD and OSA had significantly higher odds of PE compared with those with isolated OSA in adjusted models (OR 3.89; 95%CI 1.27–10.68, *p* = 0.019) [43].

Altered coagulation mechanisms in COPD affect danger for cardiovascular events as well; there is evidence linking coagulation and risk of cardiovascular disease, including the presence of several coagulation factors within the atherosclerotic lesions [44]; animal studies indicating accelerated atherosclerosis in animals with a hypercoagulable genotype [45]; associations between hypercoagulability-related hemostatic gene variants and cardiovascular disease [46]; and associations of elevated hemostatic factor levels with arterial thrombosis, cardiovascular disease stemmed from the presence of coagulation factors in atherosclerotic lesions [44].

The objective of this comprehensive review is to provide an in-depth overview of the literature regarding the prothrombotic changes generated in patients with stable COPD as well as during COPD exacerbations.

## 2. An Overview of the Coagulation Mechanism

The progress of coagulation can be generally divided into the following stages: initiation, amplification, propagation, and stabilization [47]. The mechanism of hemostasis is an intricate process which is generated through a series of clotting factors. The coagulation cascade of secondary hemostasis has two initial pathways which lead to fibrin formation and the intrinsic and the extrinsic pathway. The intrinsic pathway consists of factors I (fibrinogen), II (prothrombin), IX, X, XI, and XII. The extrinsic pathway involves factors I, II, VII, and X. The common pathway comprises factors I, II, V, VIII, X. The two pathways are a series of reactions in which coagulation factors that circulate as zymogens (inactive enzyme precursor) of a serine protease and its glycoprotein co-factors, are activated to become active components which then catalyze the next reaction in the cascade, ultimately resulting in activation of fibrinogen and formation of cross-linked fibrin. Factors II, VII, IX, X, XI and XII are serine proteases, whereas factors V, VIII are glycoproteins, and XIII is a transglutaminase. Fibrinogen, factors II, V, VII, VIII, IX, X, XI, XIII, and protein C and S are synthesized by the liver, while factor VII is created by the vascular endothelium [48].

A summary of the coagulation cascade is presented in Figure 1.

## 3. Hypercoagulability in Stable COPD

COPD is a common comorbidity or risk factor for venous thromboembolism (VTE). Both clinical and basic research has linked smoking and COPD to abnormalities of coagulation and fibrinolysis and to VTE [49]. In a 5451-patient DVT registry, in 668 individuals (12.3%) COPD was present as comorbidity [50]. 36,949 consecutive patients over 18 years with acute, symptomatic, objectively confirmed VTE were enrolled in the RIETE study. Of these, 4036 (10.9%) had COPD, of whom 2452 (61%) initially presented with PE and 1584 (39%) presented with DVT [51].

Studies conducted on stable COPD patients, have demonstrated a significant increase of coagulation factors levels. In the third National Health and Nutrition Examination Survey (NHANES III), with a total study sample size of 15,697 adults, of whom 2366 had COPD, fibrinogen levels were higher in COPD patients compared to subjects without COPD, and were associated with GOLD spirometric stages [52]. Similarly, Garcia-Rio et al., demonstrated that fibrinogen was higher in patients with COPD compared to control subjects, however fibrinogen levels were not associated with disease severity (as expressed by GOLD spirometric groups or BODE index) [53]. In the study by Eickhoff et al., fibrinogen levels were significantly higher in stable COPD patients compared to COPD-free subjects [54]. Samareh et al., demonstrated that fibrinogen levels were higher among COPD patients compared to controls; however, no statistical significance or any correlation between fibrinogen and severity of the disease was detected [55]. Agale et al., conducted a study on stable COPD patients and compared them with healthy subjects, demonstrating that fibrinogen levels are higher among COPD patients and that they correlate directly with COLD COPD groups [56]. These 5 studies demonstrate that fibrinogen levels are higher in COPD patients compared to control subjects and are associated with disease severity.

Arregui et al., observed that COPD patients had higher fibrinogen, D-dimer, factor VIII, von Willebrand factor (vWF) Ag, and vWF Ac compared with healthy individuals [57]. In another study by Zhang et al., D-dimer and fibrinogen levels were higher in stable COPD patients compared to healthy subjects [12]. In a small study, Polatli et al., examined the levels of fibrinogen, vWF, and microalbuminuria between 33 stable COPD patients, 26 patients with AECOPD and 16 control subjects. Patients with stable COPD had higher levels of fibrinogen compared to healthy subjects, while vWF and microalbuminuria did not differ significantly between the 2 groups. Patients with AECOPD had further elevated levels of fibrinogen and vWF compared to patients with stable COPD, while microalbuminuria did not differ significantly between the 2 groups [58]. These 3 studies demonstrate that COPD patients exhibit higher D-dimer, factor VIII, and vWF levels compared to control subjects, while fibrinogen, D-dimer, and vWF levels are further amplified during exacerbation.

In a study from our group, stable COPD patients had higher levels of D-dimer, fibrinogen, FII, FV, FVIII, FX, and lower levels of protein S and AT compared to COPD-free control subjects. Moreover, FII was negatively associated with FEV1 and age; FV and FX were negatively associated with FEV1; D-dimers were positively associated with pack-years and negatively associated with DLCO; fibrinogen was positively associated with CRP; FVIII was positively associated with blood neutrophil count; Protein S was positively associated with CRP and negatively associated with age and CAT score; and AT was negatively associated with blood neutrophil count, CRP and age [59]. In this study, patients with stable COPD exhibited increased levels of key coagulation factors and decreased levels of coagulation inhibitors compared to COPD-free smokers, indicating a prothrombotic state in stable COPD. Among COPD patients, FII, FV, FX, and D-dimers were associated with COPD severity; fibrinogen, FVIII, Protein S, and antithrombin were associated with inflammatory markers, whereas FII and FX were associated with LDL levels. Undas et al., detected that COPD patients had elevated fibrinogen, FII, FV, FVII, FVIII, and FIX, lower free TFP, higher total thrombin levels and higher maximum thrombin levels, compared with controls [14]. In a study with very interesting findings including 60 COPD patients with stable disease, FXIa was detected in 9 (15%) and TF activity in 7 (11.7%) COPD patients. Those patients had higher prothrombin fragment 1.2, IL-6 and CRP, thus associating COPD to hypercoagulability and chronic systemic inflammation [11]. Those 3 studies illustrated that COPD patients exhibit elevated coagulation factors and inflammatory markers compared to control subjects, featuring systemic inflammation as a possible mechanism of hypercoagulability.

Similar findings were observed for TF procoagulant activity and TAT levels in COPD patients compared to control subjects, indicating enhanced thrombin generation [60]. In a study by Ashitani et al., plasma levels of TAT complex, fibrinopeptide-A, tissue plasminogen activator-plasminogen activator inhibitor (tPA-PAI), and 3-thromboglobulin, a marker of platelet activation, were significantly higher in stable COPD patients compared to healthy never-smokers controls [8]. In the study by Szczypiorska et al., significantly higher concentrations of TF and TF pathway inhibitor (TFPI) and fibrinogen, were observed in patients with COPD. Likewise, aPTT was shortened in COPD patients compared with the control group [61]. Cella et al., compared the levels of nitric oxide (a powerful vasodilator and inhibitor of platelets aggregation), thrombomodulin (the endothelial surface site for binding and deactivating thrombin that subsequent activates protein C), and TFPI between patients with COPD and healthy controls, to evaluate the endothelial cell dysfunction associated with COPD. Both nitric oxide and thrombomodulin levels were significantly decreased in COPD patients compared to control subjects. Patients with COPD showed a significantly higher TFPI antigen plasma level than controls [62]. In the study by Waschki et al., PAI was enhanced inCOPD patients compared with COPD-free control subjects, with the highest levels observed in GOLD COPD stage II and III [63]. The 5 aforementioned studies provide evidence that TF, TFPI, TAT, tPA-PAI, and thrombomodulin levels were significantly enhanced in stable COPD patients compared to healthy controls.

## 4. Hypercoagulability during COPD Exacerbations

The risk of VTE during acute exacerbations of COPD appears to be significant [64]. In 1144 patients with COPD admitted to hospital with acute exacerbation, the VTE rate was 6.8% (78 patients). This could be underestimated due to the poor sensitivity of ultrasound for asymptomatic DVT [65]. This study excluded patients with previous thromboembolism, cancer, or heart failure, suggesting that COPD itself, or the concurrent reduced mobility, play important roles in vulnerability to VTE. Furthermore Kim et al., demonstrated that higher fibrinogen levels (>350 mg/dL) were associated with frequent exacerbations and COPD severity [66].

In the study by Saldias et al., fibrinogen levels of COPD patients were examined at stable phase and during AECOPD and were significantly higher during AECOPD. Interestingly, fibrinogen levels were reduced 15 and 30 days after AECOPD, nevertheless still remained higher than levels at stable phase [67]. In another study, fibrinogen levels of AECOPD patients were compared to convalescence values, 40 days after exacerbation, and were found to be significantly higher during AECOPD [68]. Similarly, in the study by Valipour et al., fibrinogen levels of stable COPD patients were significantly higher compared to those of healthy controls. Furthermore, fibrinogen levels of patients with AECOPD were significantly higher compared to convalescence values (6 weeks after AECOPD) and to those of the healthy controls [69]. Moreover, in the study by Wedzicha et al., including patients with COPD, the plasma fibrinogen levels were elevated during exacerbation compared to baseline values [34]. Thomas and Yuvarajan, examined fibrinogen levels in patients with stable COPD, patients with AECOPD and control subjects. Stable COPD patients exhibited higher levels compared to healthy subjects. Moreover, fibrinogen levels were further increased during exacerbation compared to stable COPD patients. Among COPD patients, fibrinogen was significantly associated with severity of obstruction [70]. These 5 studies illustrate that fibrinogen levels are elevated during AECOPD compared either to convalescence values or healthy subjects.

Maclay et al., compared the levels of fibrinogen and platelet-monocyte aggregates (a sensitive marker of platelet activation which is raised in patients with acute coronary syndromes, smokers, and in those with rheumatoid arthritis) between patients with stable COPD and healthy controls, detecting significantly higher levels among COPD patients of both fibrinogen and platelet-monocyte aggregates. D-dimer levels did not differ significantly between the two groups. Subsequently, platelet-monocyte aggregates of 12 COPD patients during AECOPD were compared to convalescence values, 2 weeks after AECOPD, being higher during AECOPD [71]. Song et al., included 60 patients with AECOPD, of which 30 had acute respiratory failure type II and 30 did not have respiratory failure. COPD patients with respiratory failure type II demonstrated higher levels of D-dimer and fibrinogen. Moreover, in the group of COPD patients with respiratory failure type II, D-dimer and fibrinogen levels had significantly positive correlations with PaCO2 and negative correlations with PO2, highlighting the contribution of hypoxia and hypercapnia to prothrombotic tendency [72].

Van der Vorm et al., demonstrated that fibrinogen, FVIII, maximum amount of thrombin (nM/min), vWF Ag, and vWF Ac, were significantly elevated during AECOPD compared to convalescence values (8 weeks after AECOPD) [73]. Similarly, in the study by Elsalam et al., including patients with AECOPD and healthy control subjects, higher levels of soluble fibrin complex, D-dimers, TAT and fibrinogen were observed in COPD patients. In further analysis, patients with severe COPD illustrated higher levels of soluble fibrin complex, D-dimer, TAT, and fibrinogen, compared to patients with moderate disease. However, there was no statistically significant difference between COPD patients and control subjects regarding the level of AT, protein C, and protein S [74]. Daga et al., studied the levels of fibrinogen, vWF Ag, PT, and aPTT between patients with AECOPD and with stable disease. Fibrinogen and vWF Ag levels were higher during AECOPD, while no difference was observed for PT and aPTT [75]. In the study by Zhang et al., levels of D-dimer and fibrinogen were elevated in COPD patients compared to controls. Among the COPD patients, D-dimer and fibrinogen levels were further increased during exacerbation compared to the stable phase [12]. These 4 studies show that during AECOPD, D-dimer, TAT, vWF, and FVIII levels are amplified compared to stable COPD values.

Husebø et al., demonstrated that during AECOPD, all markers were higher than in the stable state; TAT, APC-PCI and D-dimer. Higher D-dimer in stable COPD predicted a higher mortality (HR: 1.60 (1.24–2.05), *p* < 0.001). Higher TAT was associated with both an increased risk of later exacerbations, with a yearly incidence rate ratio of 1.19 (1.04–1.37), and a faster time to the first exacerbation (HR: 1.25 (1.10–1.42)) (*p* = 0.001, for both after adjustment) [76]. In the study by Roland et al., examining TF levels in patients with COPD, during COPD exacerbation and 4–6 weeks later, TF levels were significantly higher during COPD exacerbation, while there was no correlation observed between the rise in TF, infection symptoms, and oxygen saturation during exacerbation; treatment with antibiotics or corticosteroids and TF levels during and after COPD exacerbation [77]. These 2 studies demonstrate that TAT, APC-PCI, D-dimer, and TF levels are increased during AECOPD compared to stable COPD.

Alterations of coagulation factors and inhibitors in patients with stable COPD and during COPD exacerbations are summarized inTable 1.

## 5. Conclusions

To the best of our knowledge, this is the first comprehensive review of studies on hypercoagulability either in stable COPD or during COPD exacerbations. A large body of epidemiological data supports the hypothesis that COPD is closely linked to a hypercoagulable state. Overall, patients with stable COPD exhibit major alterations of fibrinogen, FII, FV, FVII, FVIII, FIX, D-dimers, von Willebrand factor Ag, von Willebrand factor Ac, TF, TAT, FPA, β-thromboglobulin, tPA-PAI, prothrombin fragment 1 and 2, and maximum thrombin levels, pointing to hypercoagulability. Prothrombotic state is further enhanced during exacerbations, reflected by a significant increase of TF, TAT, soluble fibrin complex, fibrinogen, D-dimers, and APC-PCI. This summary provides an in-depth overview of the alterations of coagulation factors and prothrombotic changes in COPD patients which enhances our understanding of the coexistence of cardiovascular comorbidities in these patients.

## Figures and Tables

**Figure 1 diagnostics-11-01447-f001:**
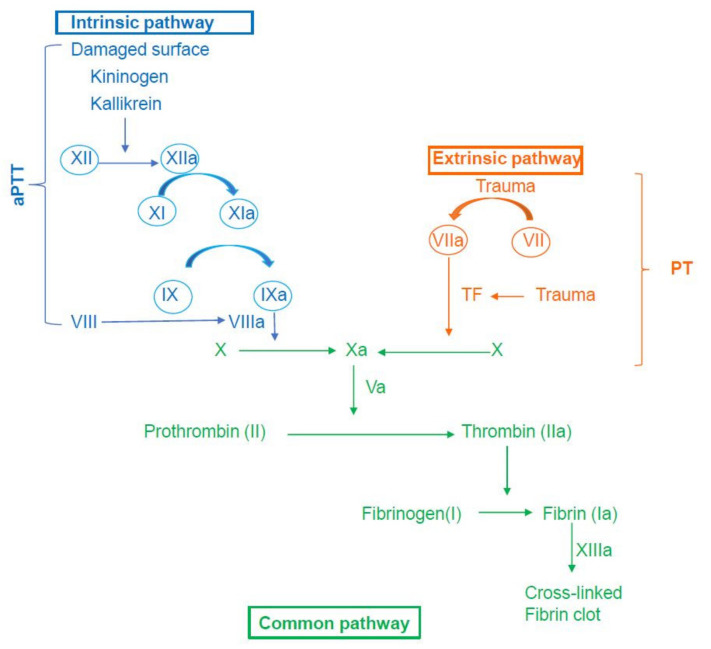
Summary of the coagulation cascade. aPTT: activated Partial Thromboplastin Time; PT: Prothrombin Time; TF: Tissue Factor.

**Table 1 diagnostics-11-01447-t001:** Alterations of coagulation factors and inhibitors in patients with stable COPD and during COPD exacerbations.

Author, Year	Participants	Disease Status	Results
Agale, 2018 [56]	50 patients with stable COPD vs. 50 healthy controls	Stable COPD	Fibrinogen (mg/dL) (455.38 ± 159.71 vs. 255.50 ± 7.98)
Arregui, 2010 [57]	51 patients with stable COPD vs. 30 healthy controls	Stable COPD	Fibrinogen (mg/dL) (469.7 vs. 334.7) D-dimers (μg/L) (339.3 vs. 244.7) Factor VIII (%) (113.7 vs. 105.1) vWF Ag (%) (113.8 vs. 110.9) vWF Ac (%) (105.9 vs. 98.8)
Ashitani, 2002 [8]	40 patients with stable COPD vs. 20 healthy controls	Stable COPD	TAT (ng/mL) (2.9 ± 1.6 vs. 1.8 ± 0.8) FPA (ng/mL) (2.7 ± 0.9 vs. 1.2 ± 0.6) tPA-PAI (ng/mL) (20.2 ± 9.3 vs. 13.8 ± 2.1) β-thromboglobulin (ng/mL) (120.0 ± 61.1 vs. 46.3 ± 8.5)
Cella, 2001 [62]	14 patients with stable COPD vs. 20 healthy controls	Stable COPD	Nitric oxide (μg/mL) (23.42 ± 1.67 vs. 40.0 ± 3.38) Thrombomodulin (ng/mL) (5.46 ± 1.32 vs. 12.9 ± 0.51) TFPI (ng/mL) (112.28 ± 6.45 vs. 77.68 ± 0.28)
Daga, 2020 [75]	30 patients with AECOPD vs. 30 stable COPD patients	AECOPD	Fibrinogen (mg/dL) (491.09 (406.86–575.31) vs. 426.86 (373.08–480.63)) vWF Ag (147.58 (141.90–153.25) vs. 128.95 (117.07–140.83)) PT (1.13 (1.09–1.18) vs. 1.09 (1.00–1.18)) ^§^ aPTT (1.14 (1.06–1.22) vs. 1.14 (1.07–1.20)) ^§^
Eickhoff, 2008 [54]	60 patients with stable COPD vs. 20 healthy controls	Stable COPD	Fibrinogen (mg/dL) (426(354–472) vs. 382 (317–428))
Elsalam, 2013 [74]	38 patients with AECOPD vs. 25 healthy controls	AECOPD	Soluble fibrin complex (μg/mL) (179.4 ± 73 vs. 3.1 ± 0.34) D-dimers (ng/mL) (444 ± 225 vs. 371 ± 72.7) TAT (ng/mL) (14.3 ± 4.0 vs. 3.6 ± 0.4) Fibrinogen (mg/dL) (338.6 ± 45 vs. 151.6 ± 4.5)
	15 patients with severe COPD vs. 10 patients with moderate COPD	AECOPD	Soluble fibrin complex (μg/mL) (231.3 ± 5 vs. 101.7 ± 34) D-dimers (ng/mL) (307.2 ± 85.9 vs. 198 ± 70) TAT (ng/mL) (17.3 ± 2.6 vs. 9.2 ± 1) Fibrinogen (mg/dL) (361.5 ± 41 vs. 297.7 ± 37)
Garcia-Rio, 2010 [53]	324 patients with stable COPD vs. 110 healthy controls	Stable COPD	Fibrinogen (mg/dL) (370 (310–410) vs. 300 (250–350))
Husebø, 2021 [76]	413 patients with stable COPD vs. 49 healthy controls	Stable COPD	TAT (ng/mL) (1.03 (0.76–1.44)) vs. (1.28 (1.04–1.49))
	148 COPD patients during AECOPD and compared to baseline values	AECOPD	TAT (ng/mL) (2.56 (1.74–3.95) vs. 1.43 (0.97–1.88)) APC-PCI (ng/mL) (489.3 (340.5–770.3) vs. 416.4 (295.5–564.3)) D-dimers (ng/mL) (763.5 (491–1192) vs. 479.7 (273–742))
Jankowski, 2011 [11]	60 stable COPD patients	Stable COPD	FXIa was detected in 9 (15%) and TF activity in 7 (11.7%) patients. Those patients had higher: ✓ prothrombin fragment 1 + 2 (398 (216) vs. 192 (42)) ✓ fibrinogen (g/L) (5.58 (2.01) vs. 3.97 (2.47))
Koutsokera, 2009 [68]	30 COPD patients during AECOPD and compared to convalescence values (40 days after AECOPD)	AECOPD	Fibrinogen (mg/dL) (545.1 ± 35.9 vs. 455.4 ± 30.0)
Kyriakopoulos, 2020 [59]	103 patients with stable COPD vs. 42 COPD-free smokers	Stable COPD	D-dimers (ng/mL) (540 ± 940 vs. 290 ± 140) Fibrinogen (mg/dL) (399 ± 82 vs. 346 ± 65) FII (%) (122 ± 22 vs. 109 ± 19) FV (%) (131 ± 25 vs. 121 ± 19) FVIII (%) (143 ± 32 vs. 122 ± 20) FX (%) (114 ± 23 vs. 100 ± 16) Protein S (%) (95.1 ± 18.74 vs. 110.5 ± 17.9) AT (%) (94.4 ± 11.5 vs. 102.3 ± 13.2)
Maclay, 2011 [71]	18 patients with stable COPD vs. 16 healthy controls	Stable COPD	Fibrinogen (g/L) (2.8 (0.5) vs. 2.7 (0.5)) Platelet-monocyte aggregates (%) (25.3 (8.3) vs. 19.5 (4.0)) D-dimers (ng/mL) (373 (137) vs. 341 (202)) ^§^
	12 COPD patients during AECOPD and compared to convalescence values (2 weeks after AECOPD)	AECOPD	Platelet-monocyte aggregates (%) (32.0 (11.0) vs. 25.5 (6.4))
Polatli, 2008 [58]	33 patients with stable COPD vs. 16 healthy controls	Stable COPD	Fibrinogen (mg/dL) (346.88 ± 92.3 vs. 289.99 ± 39.9) vWF (%) (178.26 ± 118.3 vs. 142.85 ± 57.16) ^§^ MAB (20.98 ± 28.74 vs. 10.47 ± 8.08) ^§^
	26 patients during AECOPD vs. 33 stable COPD patients	AECOPD	Fibrinogen (mg/dL) (447.67 ± 128 vs. 346.88 ± 92.3) vWF (%) (257.39 ± 157 vs. 178.26 ± 118.3) MAB (34.99 ± 46.35 vs. 20.98 ± 28.74) ^§^
Roland, 1999 [77]	30 COPD patients during AECOPD and compared to baseline values (4–6 weeks later)	AECOPD	TF (65.09 (0.00–153.80) vs. 5.92 (0.00–121.30))
Saldias, 2012 [67]	85 COPD patients during AECOPD and compared to baseline values	AECOPD	Fibrinogen (mg/dL) (395.2 ± 104.1 vs. 319.9 ± 57.1)
Samareh, 2000 [55]	31 patients with stable COPD vs. 29 healthy controls	Stable COPD	Fibrinogen (g/L) (3.81 ± 0.93 vs. 3.72 ± 0.9) ^§^
Silva, 2012 [78]	58 patients with stable COPD vs. 30 healthy controls	Stable COPD	D-dimers (ng/mL) (0.24 (0.2–0.36) vs. 0.17 (0.12–0.24)) ^§^
Song, 2013 [72]	30 AECOPD patients with ARF II vs. 30 AECOPD patients without ARF II	AECOPD	D-dimers(mg/L) (0.36 ± 0.26 vs. 0.11 ± 0.08) Fibrinogen (g/L) (4.40 ± 0.64 vs. 3.20 ± 0.64)
Szczypiorska, 2015 [61]	66 patients with stable COPD vs. 25 healthy controls	Stable COPD	TF (118.3 (27.2–373.2) vs. 83.1 (18.3–264.1)) TFPI (120.2 (52.2–323.6) vs. 96.1 (41.2–150.2)) Fibrinogen (mg/dL) (423 (323–523) vs. 359 (259–459)) aPTT (36.0 (33–39) vs. 39.8 (35–45))
Thomas, 2016 [70]	20 patients with stable COPD vs. 20 healthy controls	Stable COPD	Fibrinogen (mg/dL) (226.2 vs. 162.7)
	20 patients during AECOPD vs. 20 stable COPD patients	AECOPD	Fibrinogen (mg/dL) (275.55 vs. 226.2)
Undas, 2011 [14]	60 patients with stable COPD vs. 43 non-COPD smokers	Stable COPD	Fibrinogen (mg/dL) (414 ± 160 vs. 285 ± 55) FII (%) (115 ± 16 vs. 102 ± 10) FV (%) (114 ± 19 vs. 102 ± 12) FVII (%) (111 ± 15 vs. 102 ± 17) FVIII (%) (170 ± 34 vs. 115 ± 27) FIX (119 ± 21 vs. 107 ± 17) FX (%) (117 ± 21 vs. 110 ± 19) ^§^ Free TFPI (ng/mL) (17.7 ± 3.2 vs. 18.9 ± 3.2) Maximum thrombin levels (404 ± 76 vs. 317 ± 62) Total thrombin (92.7 ± 23.0 vs. 81.0 ± 16.5)
Vaidyula, 2009 [60]	11 patients with stable COPD vs. 10 healthy controls	Stable COPD	TF procoagulant activity (52.3 ± 5.6 vs. 20.7 ± 1.5) TAT (ng/mL) (2.99 ± 0.65 vs. 1.31 ± 0.13)
Valipour, 2008 [69]	30 patients with stable COPD vs. 30 healthy controls	Stable COPD	Fibrinogen (mg/dL) (424 (358–459) vs. 360 (326–393))
	30 COPD patients during AECOPD and compared to convalescence values (6 weeks after AECOPD)	AECOPD	Fibrinogen (mg/dL) (419 (329–470) vs. 311 (249–401))
	30 patients during AECOPD vs. 30 healthy controls	AECOPD	Fibrinogen (mg/dL) (419 (329–470) vs. 360 (326–393))
Van der Vorm, 2020 [73]	52 COPD patients during AECOPD and compared to convalescence values * (8 weeks after AECOPD)	AECOPD	Fibrinogen (g/L) (4.8 ± 1.5 vs. 4.2 ± 1.3) FVIII (%) (185 (91) vs. 153 (84)) Maximum amount of thrombin (nM/min) (1503 ± 335 vs. 1400 ± 360) vWF Ag (%) (218 (113) vs. 182 (110)) vWF Ac (%) (157 (89) vs. 135 (71))
Waschki, 2017 [63]	74 patients with stable COPD vs. 18 healthy controls	Stable COPD	Plasminogen activator inhibitor (ng/mL) (13 (10–17) vs. 11 (8–13))
Wedzicha, 2000 [34]	67 COPD patients during AECOPD and compared to baseline values	AECOPD	Fibrinogen (g/L) (4.26 ± 1.41 vs. 3.9 ± 0.67)
Zhang, 2016 [12]	43 patients with stable COPD vs. 43 healthy controls	Stable COPD	D-dimers (μg/L) (1799 (1205–2196) vs. 433 (369–456)) Fibrinogen (mg/dL) (297 ± 34.3 vs. 271 ± 66.8)
	43 COPD patients during AECOPD and compared to baseline values	AECOPD	D-dimers (μg/L) (2839 (2078–4389) vs. 1799 (1205–2196)) Fibrinogen (mg/dL) (352 ± 81.3 vs. 297 ± 34.3)

Values are presented as mean ± SD or median (IQR). ^§^ Without statistical significance. * Convalescence values available for 32 patients.

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
