# Peer review of "Hypercoagulable State in COPD-A Comprehensive Literature Review"

_diagnostics, 2021, doi:10.3390/diagnostics11081447_

Round 1

Reviewer 1 Report

I reviewed the manuscript, "Hypercoagulable state in COPD-A comprehensive literature review," with interest. This manuscript provides an overview regarding the alterations of coagulation factors and prothrombotic changes generated in patients with stable COPD and during COPD exacerbations. However, the authors should address some issues before further processing:

The authors address five possible mechanisms of the development of venous thromboembolism (VTE) and cardiovascular disease (CVD) in COPD patients (line 47). However, there is no significant discussion about these mechanisms in this manuscript. It would be helpful to discuss the associations of these mechanisms with the hypercoagulable state in COPD.

The authors provide evidence of hypercoagulability in patients with COPD during the stable and exacerbation phases. However, this manuscript does not discuss differences between the stable and exacerbation phases in the risk factors and mechanisms of the hypercoagulable state in COPD.

I feel the second paragraph ("An overview of the coagulation mechanism") is unnecessary in this manuscript. Since this is general knowledge, I do not feel the need to address it in this manuscript.

Author Response

Response to Reviewer's Comments

I reviewed the manuscript, "Hypercoagulable state in COPD-A comprehensive literature review," with interest. This manuscript provides an overview regarding the alterations of coagulation factors and prothrombotic changes generated in patients with stable COPD and during COPD exacerbations.

Response: We would like to sincerely thank the Reviewer for the effort and the time devoted while evaluating this manuscript. Your valuable comments have helped us make further corrections and involve more detailed and much needed information while preparing this revision.

Point 1: However, the authors should address some issues before further processing: The authors address four possible mechanisms of the development of venous thromboembolism (VTE) and cardiovascular disease (CVD) in COPD patients (line 47). However, there is no significant discussion about these mechanisms in this manuscript. It would be helpful to discuss the associations of these mechanisms with the hypercoagulable state in COPD.

Response 1: We acknowledge the reviewer's excellent point. We have added in the “Introduction” the following two paragraphs, analyzing the mechanisms that induce venous thromboembolism and cardiovascular disease in COPD patients (Page 2):“COPD is a disease where both local and systemic inflammation is present [17]. Prolonged systemic inflammation in COPD is indicated by increased serum concentrations of inflammatory markers such as fibrinogen, C-reactive protein (CRP), interleukin 6, interleukin 8 and TNF-a [22]. The inflammatory response and the activation of coagulation are two important mechanisms of a host's defense response against infection that do not work independently, but conjointly in an intricate and synchronous process [23]. Several investigators have suggested that low-grade inflammation in COPD largely contributes to a prothrombotic state [24]. Platelets serve a crucial role in hemostasis and inflammation [18]. Activated platelets produce several prothrombotic factors that also act as inflammatory signals, such as platelet factor 4, plasminogen activator inhibitor (PAI-1), von Willebrand factor (vWF), and fibrinogen [25]. Platelets also secrete soluble P-selectin and CD40L, which serve a dual purpose of promoting thrombus formation and activating inflammatory cells [23]. Platelet P-selectin has been shown to bind PSGL-1 on leukocytes to promote fibrin degradation and tissue factor release.  

Oxidative stress is triggered by reactive oxygen species, predisposing to thrombosis by impairing red blood cells quality and function [26], inducing endothelial dysfunction and injury, and activating leukocytes, consequently affecting the clotting system [19,27].  Oxidative stress also elevates reactive oxygen species within platelets that can augment platelet activation and thrombotic susceptibility [18]. Mechanisms by which hypoxia can stimulate a prothrombotic response include platelet activation and hypoxia-inducible transcription factors (HIFs) mediated or HIF-independent increase in procoagulant factors or reduction of coagulation inhibitors [28]. Hypoxia can also induce PAI-1 in a time-dependent process; enhance TF and repress tissue factor pathway inhibitor (TFPI) transcript expression [29], decrease protein S levels and increase thrombin generation, alterations that contribute to prothrombotic effect [30].”

Point 2: The authors provide evidence of hypercoagulability in patients with COPD during the stable and exacerbation phases. However, this manuscript does not discuss differences between the stable and exacerbation phases in the risk factors and mechanisms of the hypercoagulable state in COPD.

Response 2:Thank you very much for the very useful suggestion.We have added in the “Introduction” the following paragraph discussing the mechanisms that further amplify the the hypercoagulable state during AECOPD (Page 2-3):“COPD patients are at heightened risk of VTE during acute exacerbations of COPD (AECOPD) [31,32], which may be due to clotting system activation [33]. AECOPDs, which are crucial events in the disease’s natural history, can concomitantly lead to hypoxia and amplification of systemic inflammation, thus magnifying prothrombotic state [34]. Respiratory tract infections that are the most frequent trigger of AECOPD represent a transient risk factor for VTE [35].  In a population-based case-control study in Northern Denmark  the IRR for VTE within the first 3 months after infection was 12.5 (95% CI: 11.3-13.9) for patients with hospital-diagnosed infection and 4.0 (95% CI: 3.8-4.1) for patients treated with antibiotics in the community, compared with individuals without infection during the year before VTE [36].  Immobilization is another factor, existent during AECOPD, that imposes an even greater risk for VTE [37,38].”

Point 3: I feel the second paragraph ("An overview of the coagulation mechanism") is unnecessary in this manuscript. Since this is general knowledge, I do not feel the need to address it in this manuscript.

Response 3: We appreciate the reviewer’s suggestion. We have retained only the first part of the second paragraph and figure 1 that summarizes the coagulation cascade. Although this is also general knowledge, we believe that it serves as a useful introduction in this review that helps the reader follow the rest of the content easily.We have retained the following part:

“An overview of the coagulation mechanism

Progress of coagulation can be generally divided into the following stages: initiation, amplification, propagation, and stabilization. The mechanism of hemostasis is an intricate process that is generated through a series of clotting factors. The coagulation cascade of secondary hemostasis has two initial pathways which lead to fibrin formation, the intrinsic and the extrinsic pathway. The intrinsic pathway consists of factors I (fibrinogen), II (prothrombin), IX, X, XI, and XII. The extrinsic pathway involves factors I, II, VII, and X. The common pathway comprises factors I, II, V, VIII, X. The two pathways are a series of reactions, in which coagulation factors that circulate as zymogens (inactive enzyme precursor) of a serine protease and its glycoprotein co-factors are activated to become active components that then catalyze the next reaction in the cascade, ultimately resulting in activation of fibrinogen and formation of cross-linked fibrin. Factors II, VII, IX, X, XI and XII are serine proteases, whereas factors V, VIII are glycoproteins, and XIII is a transglutaminase. Fibrinogen, factors II, V, VII, VIII, IX, X, XI, XIII, and protein C and S are synthesized by the liver, while factor VII is created by the vascular endothelium. A summary of the coagulation cascade is presented on figure 1”.

Reviewer 2 Report

In the present study „Hypercoagulable state in COPD - A comprehensive literature review“ the authors have reviewed recent literature on alterations of coagulation factors and prothrombotic changes generated in patients with stable COPD and during COPD exacerbations. The manuscript is very comprehensive, concise, and written in a nice way. However, from my point of view, I would say that the Authors should take another look at the last two parts of the manuscript. The parts of the manuscript „Hypercoagulability in stable COPD“ and „Hypercoagulability during exacerbations“ are informative and lots of recent literature was presented, however, it seems that all of the textual parts is presented in the Table format, and I would suggest that it should stay like this. I think that the textual part of the manuscript, related to these topics is hard to follow, with too much information and without a particular conclusion. In my opinion, this part of the manuscript should be processed in a way to comment on the specific parameters/biomarkers and how they can be used in the context of COPD and VTE.

I am not an expert in English, but it may be worth reading the text by a native speaker. For me, it looks fine!

Author Response

Response to Reviewer's Comments

In the present study “Hypercoagulable state in COPD - A comprehensive literature review “the authors have reviewed recent literature on alterations of coagulation factors and prothrombotic changes generated in patients with stable COPD and during COPD exacerbations. The manuscript is very comprehensive, concise, and written in a nice way.

Response: We would like to sincerely thank the Reviewer for the effort and the time devoted while evaluating this manuscript. Your valuable comments have helped us make further corrections and involve more detailed and much needed information while preparing this revision.

Point 1: However, from my point of view, I would say that the Authors should take another look at the last two parts of the manuscript. The parts of the manuscript “Hypercoagulability in stable COPD“ and “Hypercoagulability during exacerbations“ are informative and lots of recent literature was presented, however, it seems that all of the textual parts is presented in the Table format, and I would suggest that it should stay like this. I think that the textual part of the manuscript, related to these topics is hard to follow, with too much information and without a particular conclusion. In my opinion, this part of the manuscript should be processed in a way to comment on the specific parameters/biomarkers and how they can be used in the context of COPD and VTE.

Response 1: Thank you for the very useful and practical suggestion. We have modified the text by reducing its length, removing all the results that are presented in the Table and commenting on the specific parameters/biomarkers and how they can be used in the context of COPD and VTE as follows:

“Hypercoagulability in stable COPD

   COPD is a common comorbidity or risk factor for venous thromboembolism (VTE). Both clinical and basic researches have linked smoking and COPD to abnormalities of coagulation and fibrinolysis and to VTE [49]. In a 5,451-patient DVT registry, in 668 individuals (12.3%)  COPD was present as comorbidity [50]. 36,949 consecutive patients over 18 years with acute, symptomatic, objectively confirmed VTE were enrolled in the RIETE study. Of them, 4,036 (10.9%) had COPD of whom 2,452 (61%) initially presented with PE and 1,584 (39%) presented with DVT [51].

    Studies held on stable COPD patients, have demonstrated a significant increase of coagulation factors levels. In the third National Health and Nutrition Examination Survey (NHANES III), with a total study sample size of 15,697 adults, out of whom 2,366 had COPD, fibrinogen levels were higher in COPD patients compared to subjects without COPD, and were associated with GOLD spirometric stages [52]. Similarly, Garcia-Rio et al. demonstrated that fibrinogen was higher in patients with COPD compared to control subjects, however fibrinogen levels were not associated with disease severity (as expressed by GOLD spirometric groups or BODE index) [53]. In the study by Eickhoff et al. fibrinogen levels were significantly higher in stable COPD patients compared to COPD-free subjects [54]. Samareh et al. exhibited that fibrinogen levels were higher among COPD patients compared to controls, however, not achieving statistical significance and without any correlation between fibrinogen and severity of the disease, being detected [55]. Agale et al., conducted a study on stable COPD patients and compared them with healthy subjects, demonstrating that fibrinogen levels are higher among COPD patients and that they correlate directly with COLD COPD groups [56]. These 5 studies demonstrate that fibrinogen levels are higher in COPD patients compared to control subjects and are associated with disease severity.  

   Arregui  et al. observed that COPD patients had higher fibrinogen, D-dimer, factor VIII, von Willebrand factor (vWF) Ag and vWF Ac compared with healthy individuals [57]. In another study by Zhang et al. D-dimer and fibrinogen levels were higher in stable COPD patients compared to healthy subjects [12]. In a small study, Polatli et al. examined the levels of fibrinogen, vWF and microalbuminuria between 33 stable COPD patients, 26 patients with AECOPD and 16 control subjects. Patients with stable COPD had higher levels of fibrinogen compared to healthy subjects, while vWF and microalbuminuria did not differ significantly between the 2 groups. Patients with AECOPD had further elevated levels of fibrinogen and vWF compared to patients with stable COPD, while microalbuminuria did not differ significantly between the 2 groups [58]. These 3 studies demonstrate that COPD patients exhibit higher D-dimer, factor VIII, and vWF levels compared to control subjects, while fibrinogen, D-dimer, and vWF levels are further amplified during exacerbation.

    In a study from our group, stable COPD patients had higher levels of D-dimer, fibrinogen, FII, FV, FVIII, FX and lower levels of protein S and AT compared to COPD-free control subjects. Moreover, FII was negatively associated with FEV1 and age; FV and FX were negatively associated with FEV1; D-dimers were positively associated with pack-years and negatively associated with DLCO; fibrinogen was positively associated with CRP; FVIII was positively associated with blood neutrophil count; Protein S was positively associated with CRP and negatively associated with age and CAT score; and AT was negatively associated with blood neutrophil count, CRP and age [59]. In this study, patients with stable COPD exhibited increased levels of key coagulation factors and decreased levels of coagulation inhibitors compared to COPD-free smokers, indicating a prothrombotic state in stable COPD. Among COPD patients, FII, FV, FX and D-dimers were associated with COPD severity; fibrinogen, FVIII, Protein S and antithrombin were associated with inflammatory markers; whereas FII and FX were associated with LDL levels. Undas et al. detected that COPD patients had elevated fibrinogen, FII, FV, FVII, FVIII, and FIX, lower free TFP), higher total thrombin levels and higher maximum thrombin levels, compared with controls [14]. In a study with really interesting findings including 60 COPD patients with stable disease, FXIa was detected in 9 (15%) and TF activity in 7 (11.7%) COPD patients. Those patients had higher prothrombin fragment 1.2, IL-6 and CRP, thus associating COPD to hypercoagulability and chronic systemic inflammation [11]. Those 3 studies illustrated that COPD patients exhibit elevated coagulation factors and inflammatory markers compared to control subjects, featuring systemic inflammation as a possible mechanism of hypercoagulability.

    Similar findings were observed for TF procoagulant activity and TAT levels in COPD patients compared to control subjects, indicating enhanced thrombin generation [60]. In a study by Ashitani et al. plasma levels of TAT complex, fibrinopeptide-A, tissue plasminogen activator-plasminogen activator inhibitor (tPA-PAI), and 3-thromboglobulin, a marker of platelet activation, were significantly higher in  stable COPD patients compared to  healthy never-smokers controls [8]. In the study by Szczypiorska et al. significantly higher concentrations of TF and TF pathway inhibitor (TFPI) and fibrinogen, were observed in 66 patients with COPD. Likewise, aPTT was shortened in COPD patients compared with the control group [61]. Cella et al. compared the levels of nitric oxide (powerful vasodilator and inhibitor of platelets aggregation), thrombomodulin (the endothelial surface site for binding and inactivating thrombin that subsequent activates protein C), and TFPI between patients with COPD and healthy controls, to evaluate the endothelial cell dysfunction associated with COPD. Both nitric oxide and thrombomodulin levels were significantly decreased in COPD patients compared to control subjects. Patients with COPD showed a significantly higher TFPI antigen plasma level than controls [62]. In the study by Waschki et al PAI was enhanced in COPD patients compared with COPD-free control subjects, with the highest levels observed in GOLD COPD stage II and III [63]. The 5 aforementioned studies provide evidence that TF, TFPI, TAT, tPA-PAI, and thrombomodulin levels were significantly enhanced in stable COPD patients compared to healthy controls.

Hypercoagulability during COPD exacerbations

    The risk of VTE during acute exacerbations of COPD appears to be significant [64]. In 1144 patients with COPD admitted to hospital with acute exacerbation, the VTE rate was 6.8% (78 patients). This could be underestimated due to the poor sensitivity of ultrasound for asymptomatic DVT [65]. This study excluded patients with previous thromboembolism, cancer, or heart failure, suggesting that COPD itself or the concurrent reduced mobility play important roles in the vulnerability to VTE. Furthermore Kim et al. exhibited that higher fibrinogen levels (>350 mg/dl) were associated with frequent exacerbations and COPD severity [66]. 

    In the study by Saldias et al. fibrinogen levels of 85 COPD patients were examined at stable phase and during AECOPD, and were significantly higher during AECOPD. Interestingly, fibrinogen levels were reduced 15 and 30 days after AECOPD, nevertheless still remained higher than levels at stable phase [67]. In another study, fibrinogen levels of 30 AECOPD patients were compared to convalescence values, 40 days after exacerbation, and were found to be significantly higher during AECOPD [68]. Similarly, in the study by Valipour et al. fibrinogen levels of stable COPD patients were significantly higher compared to those of healthy controls. Furthemore, fibrinogen levels of 30 patients with AECOPD were significantly higher compared to convalescence values (6 weeks after AECOPD) and to those of the 30 healthy controls [69]. Moreover, in the study by Wedzicha et al. including 67 patients with COPD, the plasma fibrinogen levels were elevated during exacerbation compared to baseline values [34]. Thomas and Yuvarajan, examined fibrinogen levels in patients with stable COPD, patients with AECOPD and control subjects. Stable COPD patients exhibited higher levels compared to healthy subjects. Moreover fibrinogen levels were further increased during exacerbation compared to stable COPD patients. Among COPD patients, fibrinogen was significantly associated with severity of obstruction [70]. These 5 studies illustrate that fibrinogen levels are elevated during AECOPD compared either to convalescence values or healthy subjects.

   Maclay et al. compared the levels of fibrinogen and platelet-monocyte aggregates (a sensitive marker of platelet activation that are raised in patients with acute coronary syndromes, smokers and in those with rheumatoid arthritis) between patients with stable COPD and  healthy controls, detecting significantly higher levels among COPD patients of both  fibrinogen and platelet-monocyte aggregates. D-dimer levels did not differ significantly between the two groups. Subsequently platelet-monocyte aggregates of 12 COPD patients during AECOPD were compared to convalescence values, 2 weeks after AECOPD, being higher during AECOPD [71]. Song et al. included 60 patients with AECOPD, 30 of which had acute respiratory failure type II and 30 did not have respiratory failure. COPD patients with respiratory failure type II demonstrated higher levels of D-dimer and fibrinogen. Moreover, in the group of COPD patients with respiratory failure type II, D-dimer and fibrinogen levels had significantly positive correlations with PaCO2 and negative correlations with PO2, highlighting the contribution of hypoxia and hypercapnia to prothrombotic tendency [72].

    Van der Vorm et al. demonstrated that fibrinogen, FVIII, maximum amount of thrombin (nM/min), vWF Ag, and vWF Ac, were significantly elevated during AECOPD compared to convalescence values (8 weeks after AECOPD) [73]. Similarly, in the study by Elsalam et al. including patients with AECOPD and healthy control subjects, higher levels of soluble fibrin complex, D-dimers, TAT and fibrinogen were observed in COPD patients. In further analysis patients with severe COPD illustrated higher levels of soluble fibrin complex, D-dimer, TAT and fibrinogen, compared to patients with moderate disease. However there was no statistically significant difference between COPD patients and control subjects regarding the level of AT, protein C and protein S [74]. Daga et al. studied the levels of fibrinogen, vWF Ag, PT and aPTT between patients with AECOPD and with stable disease. Fibrinogen and vWF Ag levels were higher during AECOPD, while no difference was observed for PT and aPTT [75]. In the study by Zhang et al. levels of D-dimer and fibrinogen were elevated in COPD patients compared to controls. Among the COPD patients, D-dimer and fibrinogen levels were further increased during exacerbation compared to the stable phase [12]. These 4 studies exhibit that during AECOPD, D-dimer, TAT, vWF, and FVIII levels are amplified compared to stable COPD values.

Husebø et al. demonstrated that during AECOPD, all markers were higher than in the stable state; TAT, APC-PCI and D-dimer. Higher D-dimer in stable COPD predicted a higher mortality (HR: 1.60 (1.24-2.05), p<0.001). Higher TAT was associated with both an increased risk of later exacerbations, with a yearly incidence rate ratio of 1.19 (1.04-1.37), and a faster time to the first exacerbation (HR: 1.25 (1.10-1.42)) (p=0.001, for both after adjustment) [76]. In the study by Roland et al. examining TF levels in patients with COPD, during COPD exacerbation and 4-6 weeks later, TF levels were significantly higher during COPD exacerbation, while there was no correlation observed between the rise in TF, infection symptoms, and oxygen saturation during exacerbation; treatment with antibiotics or corticosteroids and TF levels during and after COPD exacerbation [77]. These 2 studies demonstrate that TAT, APC-PCI, D-dimer, and TF levels are increased during AECOPD compared to stable COPD.

Alterations of coagulation factors and inhibitors in patients with stable COPD and during COPD exacerbations are summarized in Table 1.

Point 2: I am not an expert in English, but it may be worth reading the text by a native speaker. For me, it looks fine!

Response 2: Thank you for your comment. The text was edited by a native English speaker, performing all the necessary grammatic and syntax changes.

Round 2

Reviewer 1 Report

I have no further comments, the authors have successfully addressed all previous concerns.